# Organic Fertilization with Biofertilizer Alters the Physical and Chemical Characteristics of Young Cladodes of *Opuntia stricta* (Haw.) Haw.

Márcio S. Silva [1,*] , Jackson S. Nóbrega [1] , Cleberton C. Santos [2] , Franciscleudo B. Costa [1] , Daniel C. Abreu [3] , Wininton M. Silva [3,4] , Aaron Kinyu Hoshide [3,5] , Fernando A. L. Gomes [6] , Ulisses S. Pereira [1] , Jéssica A. Linné [2] and Silvana P. Q. Scalon [2]

[1] Center of Science and Agri-Food Technology, Federal University of Campina Grande, Street Jario Vieira Feitosa, 1770, Pombal 58840-000, PB, Brazil
[2] Faculty of Agricultural Science, Federal University of Grande Dourados, Highway Dourados, Itahum/km 1, Dourados 79804-970, MS, Brazil
[3] AgriSciences, Universidade Federal do Mato Grosso, Caixa Postal 729, Sinop 78550-970, MT, Brazil
[4] Empresa Mato-Grossense de Pesquisa, Assistência e Extensão Rural (EMPAER-MT), Centro Político Administrativo, Cuiabá 78049-903, MT, Brazil
[5] College of Natural Sciences, Forestry and Agriculture, The University of Maine, Orono, ME 04469, USA
[6] Department of Plant Science and Environmental Science, Federal University of Paraíba, Unit II, Areia 58397-000, PB, Brazil
* Correspondence: marcyyo@outlook.com; Tel.: +55-(067)-999447958

**Abstract:** Cactus cladodes are consumed by humans in arid and semiarid regions of the world. The use of biofertilizers when cultivating cacti can improve the physical and chemical characteristics of the soil, as well as the cladodes' productivity and physical-chemical quality. We evaluated the physical and physical-chemical qualities of different lengths of *Opuntia stricta* (Haw.) Haw. Cladodes were grown with different biofertilizer doses. The $3 \times 5$ factorial design employed corresponded to three cladode sizes (8–12, 12–16, and 16–20 cm) and five doses of biofertilizer (0, 5, 10, 15, and 20%) with three repetitions in a completely randomized design. Cladode characteristics were evaluated 40 days after emergence: diameter, fresh mass, soluble solids, pH, titratable acidity, soluble solid and titratable acidity ratio (SS/TA), ascorbic acid, phenolic compounds, total soluble sugars, chlorophyll *a*, *b*, and total, carotenoids, and respiration. The *Opuntia stricta* cladodes sized 16–20 cm exhibited better physical and physical-chemical qualities as well as better respiratory rates. The biofertilizer improved the cladodes' physical and physical-chemical qualities, regardless of the cladode's size. *Opuntia stricta* (Haw.) Haw. cladodes had levels of antioxidant compounds similar to those of some conventional vegetables, making them suitable for improving human health in arid environments.

**Keywords:** human food; *Opuntia stricta*; organic production; non-conventional vegetable; soluble solids

## 1. Introduction

Cacti (*Opuntia* spp.) are species that belong to the Cactaceae family. They are abundantly grown in Brazil's northeastern semiarid region and employed as a source of human and animal nutrition [1,2]. Within this cactus genus, *Opuntia tuna* (L.) Mill is highlighted for presenting resistance against the prickly pear cochineal and for yielding satisfactory productions in low-fertility soils [3]. Another species, *Opuntia stricta* (Haw.) Haw. is used in human nutrition due to its nutraceutical properties. Consequently, young cladodes are consumed in juices or dehydrated powders with high fiber contents [4]. Its use has become viable due to its low acquisition cost and because its constitution presents mineral elements, phenolic compounds, vitamins, proteins, and antioxidant carbohydrate compounds [5–7].

Besides these characteristics, cladodes are employed for phytotherapeutic purposes to treat gastritis, hyperglycemia, diabetes, arteriosclerosis, and prostatic hypertrophy [8].

To obtain quality cladodes, alternative techniques can be adopted to reduce production costs. One such technique is to use biofertilizers, which can increase yield by acting on the accumulation of solutes and the adjustment of vegetable metabolism, which increases biomass production [9]. Furthermore, biofertilizers have organic substances that can improve the soil's properties and increase plant growth by satisfying the cultivar's needs for both macro- and micro-nutrients as well as organic matter [10,11]. Therefore, the use of biofertilizers can increase the production of cactus cladodes, improve the chemical, physical, and biological characteristics of the soil, control pests and diseases, and improve the physical-chemical qualities of different vegetables [12].

Currently, the Northeast region of Brazil has the largest cultivated area of forage palm in the world, reaching around 550,000 hectares, with the planted area in Brazil being approximately 600,000 hectares [12,13]. In arid and semi-arid regions, such as those found in Brazil, cactus pear is clearly important as a source of animal feed and is used as a strategy for coping with drought. The main factor(s) that encourage or discourage prominent cultivation of *Opuntia stricta* (Haw.) Haw. are the supply of food alternatives to these cacti's cladodes and fruits for (1) foraging for animals with high nutritional quality and (2) human food that can guarantee both food security and nutritional quality [14].

Cactus use in human food is widespread in some parts of the world, such as Mexico [4]. In Brazil, the use of young *Opuntia* spp. cladodes is still relatively new, requiring the development of technologies for the production of cladodes with higher nutritional quality. In general, the management of this cactus is low-intensity, with organic fertilization being a viable alternative for producers in arid climates. Efficient use of organic fertilizers such as animal manures can reduce production costs and improve the nutritional qualities of cactus cladodes intended for human consumption [1].

In our research, we hypothesize that increased biofertilizer availability alters the physical and physicochemical quality of *Opuntia stricta* at different growth stages. We expect that there is an optimal amount of biofertilizer that should be added to maximize the size and quality of *Opuntia stricta* (Haw.) Haw. cactus cladodes for human consumption. In this context, the objective of our research was to evaluate the physical and physical-chemical qualities of different lengths of *Opuntia stricta* cladodes grown with different doses of biofertilizer.

## 2. Materials and Methods

### 2.1. Plant Material and Cultivation Conditions

The mature *Opuntia stricta* plants used to propagate the plants for our study were obtained from an experimental area of the Center of Agri-food Science and Technology (CCTA) of the Federal University of Campina Grande (UFCG), Pombal, Paraíba, Brazil, located at 6°48′16′′ of south latitude and 37°49′15′′ of west longitude, at an altitude of 175 m. The mature *Opuntia stricta* cladodes were planted in the CCTA's Experimental Farm Professor Rolando Enrique Rivas Castellón, which belongs to the UFCG, in São Domingos, Paraíba, Brazil, located at 6°48′45′′ of south latitude and 37°55′43′′ of west longitude, at an altitude of 190 m, around 38 km from Pombal, Paraíba. The predominant climate of the region is of type BSh (Köppen), a hot semiarid with an average annual precipitation of 750 mm and concentrated rainfalls from the months of December to April [15]. The temperature and relative humidity are 36 °C and 20%, respectively.

The experiment was installed in June 2018, using 400 cladodes of *O. stricta*, harvested completely mature at 19 months of age, as propagation material. These cuttings were dried in the shade for six days for healing using an application of Bordeaux mixture to prevent fungal diseases and control pests. The experimental design used at the research site and during laboratory analyses was the Completely Randomized Design (CRD), in a factorial design of 3 × 5, with 3 repetitions for each treatment, that is, three cladode size groups: 8–12, 12–16, and 16–20 cm (Figure 1), and five biofertilizer doses (0, 5, 10, 15, and

20%). The cladodes were planted in blocks with an area of 342 m$^2$ in the vertical position and in the east–west direction. The CRD consisted of four blocks, with each block being 19 m × 9 m = 171 m$^2$, containing 5 experimental patches. The patches were composed of four plantation lines with five plants spaced 0.5 m apart, totaling 20 experimental patches with 4 repetitions. The space between lines, patches, and blocks was 1.6, 1.6, and 2.0 m, respectively.

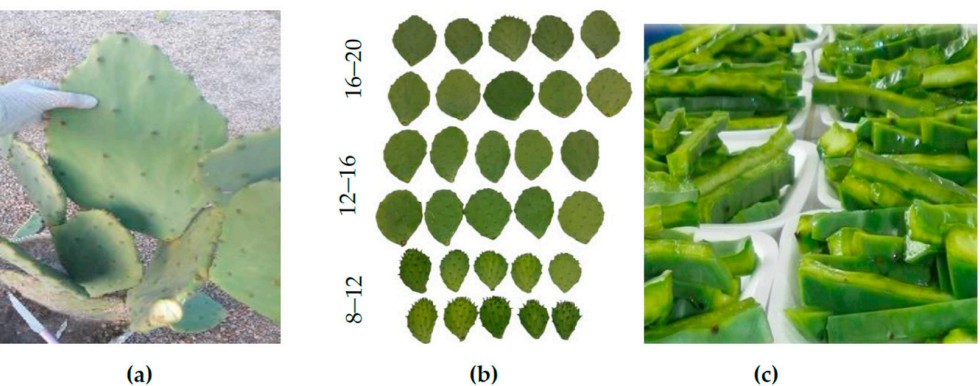

**Figure 1.** (**a**) *Opuntia stricta* (Haw.) Haw. plant evaluated for different biofertilizer applications; (**b**) cladodes with 8 to 12, 12 to 16, and 16 to 20 cm; and (**c**) *Opuntia stricta* minimally processed for human consumption. Source: the authors.

When planting, about two-thirds of each cladode was buried in the soil. Irrigation was performed twice a week for 30 min at the end of the day through a drip system with a flow of 2.2 L per hour, with drip lines spaced 0.5 m apart. The cactus plantings were thinned three times within 107 days after planting (22 September2018, 3 November 2018, and 15 December 2018), after which only the main cladodes were left for the evaluation of the cladodes' growth. After the second thinning, biofertilizer doses of 0% (just water), 5%, 10%, 15%, and 20% were applied to the mature cladodes. The biofertilizer used had a macronutrient analysis for nitrogen (N), phosphorus (P), and potassium (K) shown in Table 1. The applications were performed every 20 days with a volume of 312.5 mL. The control of spontaneous vegetation was performed through weeding.

**Table 1.** Physical and chemical characteristics of the biofertilizer used for the cultivation of *Opuntia stricta* (Haw.) Haw.

| Sample | Chemical and Physical Characteristics * | | | | | | | | |
|---|---|---|---|---|---|---|---|---|---|
| | N | P | K$^+$ | Ca$^{2+}$ | Mg$^{2+}$ | Na | OM | EC | pH |
| | % | mg/dm$^3$ | | | mg/dm$^3$ | mg/dm$^3$ | g/dm$^3$ | dS/m | (H$_2$O) |
| Biofertilizer | 0.2 | 546.6 | 8844.1 | 3.3 | 1.1 | 5.54 | 79.4 | 7.9 | 9.7 |

\* Measured characteristics included percent (%) nitrogen (N), phosphorus (P), potassium (K$^+$), calcium (Ca$^{2+}$), magnesium (Mg$^{2+}$), sodium (Na), and organic matter (OM) measured in milligrams (mg) per cubic decimeter (dm$^3$). Electric conductivity (EC) was measured in deciSiemens (dS) per meter (m). The pH was measured in aqueous extract (1:2.5).

The material harvested after the second thinning and after the application of the treatments had its physical and physical-chemical features analyzed. The plants were evaluated in the Laboratory of Food Analysis, Chemistry and Biochemistry of Foods at the Universidade Federal de Campina Grande's Pombal campus, Paraiba state, Brazil. Laboratory samples were taken from plants in the central six of each patch's middle line; plants along the borders were not sampled.

The biofertilizer used was prepared as follows: Ten kilograms of cattle manure were added to 40 L of water in a plastic container with a volume of 100 L, after which it was fermented for 30 days and agitated once a week. The first and second application of the

biofertilizer happened on 3 November 2018 and 23 November 2018, 149 and 169 days after planting, respectively. The physical and chemical characteristics of the biofertilizer are presented in Table 1.

The characteristics of the soil used in our study is of the Eutrophic Fluvic Neosol (RYve) type (Table 2), sampled from the CCTA/UFCG experimental farm [16]. Fluvial Neosols (RY) are non-hydromorphic mineral soils, originating from recent sediments from the Quaternary period. They are formed by overlapping layers of recent alluvial sediments without pedogenetic relationships between them, due to their low pedogenetic development. Generally, they present very diversified thickness and granulometry, along the soil profile, due to the diversity and deposition forms of the original material: Fluvisols - BRAZILIAN SYSTEM Fluvisols - WRB/FAO Entisols (Fluvents) - Soil Taxonomy [17]

**Table 2.** Physical and chemical characteristics of the Eutrophic Fluvic Neosol (RYve) soil used in the experimental area for the production of *Opuntia stricta* (Haw.) Haw.

| Soil Type of Eutrophic Fluvic Neosol (RYve) | Chemical and Physical Characteristics * | | | | | | | | |
|---|---|---|---|---|---|---|---|---|---|
| | P | $K^+$ | $Ca^{2+}$ | $Mg^{2+}$ | Na | SB | CEC | OM | pH |
| | mg/dm$^3$ | | cmolc/dm$^3$ | | | | | g/kg | (H$_2$O) |
| 0–0.2 m depth | 148.9 | 263.7 | 3.0 | 1.34 | 0.07 | 5.09 | 6.42 | 7.1 | 7.6 |

* Measured characteristics included percent (%) nitrogen (N), phosphorus (P), potassium ($K^+$), calcium ($Ca^{2+}$), magnesium ($Mg^{2+}$), sodium (Na), SB (sum of bases), CEC (cation exchange capacity), and organic matter (OM), measured in milligrams (mg) per cubic decimeter (dm$^3$). Electric conductivity (EC) was measured in deciSiemens (dS) per meter (m). The pH was measured in aqueous extract (1:2.5). Soil sampled at a depth of 0 to 0.2 m.

### 2.2. Physical and Physical-Chemical Analyses

The longitudinal and transversal diameters of the cladode were calculated with a digital caliper. Cladode dimensions were measured in centimeters (cm). The fresh mass measured in grams (g) was determined by weighting the young cladodes on a semi-analytical scale with a precision of 0.01 g.

Physico-chemical variables of soluable solids, pH, titratable acidity, ascorbic acid, phenolic compounds, total soluble sugars, chlorophyll and carotenoids, and respiration rate of palm cladodes were evaluated. Chemical characteristics such as soluble solids of the young cladode extract were determined in a digital refractometer (ITREFD65) with automatic temperature compensation. The results were expressed in percentages. Power of Hydrogen (pH) of the young cladode extracts was determined with a benchtop digital potentiometer (Digimed, model DM-22) [18]. The $H^+$ ion concentration, measured in micromolar (μM), required a direct reading of the pulp in digital potentiometer and calculated according to the equation: pH = log [$H^+$]. To measure titratable acidity, one gram of the young cactus cladode extract was added to 50 mL of distilled water. The solution was titrated with sodium hydroxide (NaOH) at 0.1 molar (M) until the endpoint of the phenolphthalein indicator, as confirmed by the pH range of 8.2 [18]. The total titratable acidity was expressed as percentage of malic acid. The soluble solid and titratable acidity ratio (SS/TA) was calculated by dividing the soluble solid values by the titratable acidity values.

### 2.3. Nutritional Analyses

Nutritional characteristics of cactus cladodes included ascorbic acid, which was measured by taking one gram of the young cactus cladode extract and adding to 49 mL of oxalic acid at 0.5% concentration. This was then titrated with Tillmans' solution until a pink color was obtained, according to the method (365/IV) described by Institute [18]. Soluble phenolic compounds were estimated through the Folin-Ciocalteu method [19]. Water and the Folin-Ciocalteu reagent were added to the sample, followed by agitation and rest for 5 min. After the reaction time, 250 microliter (μL) of sodium carbonate was added,

followed by more agitation, and then rest in a water bath at 40 °C for 30 min. The sample was cooled, and the reading was performed at 765 nm in a spectrophotometer.

Other nutritional characteristics included total soluble sugars (%) determined through the Anthrone method [20]. The extract was obtained by the dilution of 1 g of cactus pulp in 10 mL of distilled water. The samples were prepared in ice bath through the addition, in a tube, of 0.5 mL of the extract, 0.5 mL of distilled water, and 2 mL of the anthrone solution (0.2%), followed by agitation and rest in a thermostatic bath at 100 °C for 3 min. The samples' reading was performed in a spectrophotometer at 620 nm, employing glucose as reference to obtain the standard curve.

Chlorophyll and carotenoid levels were determined as described by Lichtenthaler [21]. Two grams (2.0 g) of the cladodes' cellular extract were weighed and ground in a mortar, after which 0.2 g of calcium carbonate ($CaCO_3$) and about 3 mL of acetone at 80% were added to the mortar. After the extract's transition, it was transferred into a centrifuge tube. Then, the leftover in the mortar was washed with 2 mL of acetone at 80% and added to the tubes. They were centrifuged for 10 min at 10 °C and 3000 revolutions per minute (rpm). An aliquot of the supernatant was poured into a cuvette. The samples' readings were performed in a spectrophotometer at the wavelengths of 470.646, and 663 nm.

### 2.4. Respiratory Rate

The respiratory rate of cactus cladodes was measured in milligrams of carbon dioxide ($CO_2$)/kg/hour was determined according to [22] and adaptations described by [23]. The *Opuntia stricta* cladodes were stored in 750 mL plastic pots with covers for 6 h, remaining over a bench at a controlled room temperature of 24 ± 1 °C and 32 ± 2% relative humidity (RH). Another container holding NaOH 0.5 N was placed in the aforementioned containers. They worked as fixers of the $CO_2$ generated in the respiration process. In the experiment, 0.5 mL of NaOH 0.5 N was employed, containing five repetitions, one of which was the sample called white test (repetition prepared without the cladode). To avoid gas exchange with the environment, the containers' covers were wrapped with a silicon layer. After the 6-h period, the NaOH solution was removed from the container and received three drops of the phenolphthalein indicator and 10 mL of $BaCl_2$ 0.2 N in an Erlenmeyer flask before being submitted to titration with hydrochloric acid at 0.1 N. The final calculation of the respiratory rate, at each analysis time, was based on the repetitions, whose result was expressed in mg $CO_2$/gram of fresh mass.

### 2.5. Statistical Analyses

Statistical analyses comparing all physical-chemical contrasts were performed using the Sisvar® program [24]. The final data were submitted for analysis of variance at 5% probability through the F-test. In cases of significant effects, Tukey's test at 5% ($p > 0.05$) was used when comparing cladodes of different sizes. Non-linear regression analysis was used to model functional relationships between each physical-chemical characteristic and biofertilizer doses (0—just water, 5, 10, 15, and 20%).

## 3. Results

The interaction between the factors was significant for the cladodes' thickness and fresh mass. The isolated effects were significant for the biofertilizer doses and cladode sizes (Table 3). Organic fertilization via biofertilizer promotes improvements in the physical attributes of the palm cladodes; it was observed that the thickness of the cladodes at 16–20 cm (cm) was greater (0.80 cm) at the biofertilizer level of 10%. As for the cladodes of 8–12 and 12–16 cm, they presented their maximum increases (0.60 and 0.73 cm) at the doses of 10 and 20%, respectively (Figure 2a). The greatest fresh mass of the cladodes of 16–20, 8–12, and 12–16 cm was seen at the biofertilizer dose of 20%, with averages of 143.55, 29.52, and 121.99 g, respectively (Figure 2b).

**Table 3.** Summary of analysis of variance for longitudinal diameter, transverse diameter, thickness, and fresh mass of palm cladodes *Opuntia stricta* (Haw.) Haw. for different biofertilizer doses and cactus cladode sizes.

| Variation Sources | Degrees of Freedom | Medium squares | | | |
|---|---|---|---|---|---|
| | | Longitudinal Diameter | Transverse Diameter | Thickness | Fresh Mass |
| Doses (D) | 4 | 262.9 ** | 106.02 ** | 0.013 ** | 32,118.6 ** |
| Length (C) | 2 | 1.24 * | 10.76 ** | 0.12 ** | 1907.9 ** |
| D × C | 8 | 0.77 ns | 2.08 ns | 0.0059 * | 608.3 * |
| Residue | 28 | 0.38 | 2.17 | 0.0022 | 219.6 |
| Average | | 14.94 | 10.85 | 0.63 | 75.6 |
| Coefficient of Variation (%) | | 4.14 | 13.6 | 7.53 | 19.6 |

The symbol ns denotes differences between factors are not significant, while ** and * indicate significant differences at 1% and 5% probability, respectively, using the F-test.

The longitudinal (19.10 cm) and transversal (13.28 cm) diameters of the cladodes with 16–20 cm were greater in comparison with the cladodes of 12–16 and 8–12 cm, whose values were (14.80 and 11.28 cm) and (10.70 and 8.01 cm), respectively (Figure 3a,b). Biofertilization also increased the growth of cladodes, with gains in longitudinal diameter (15.4 cm) and transversal diameter (12.2 cm) that were superior in the cladodes that received a biofertilizer dose of 20% (Figure 4a,b).

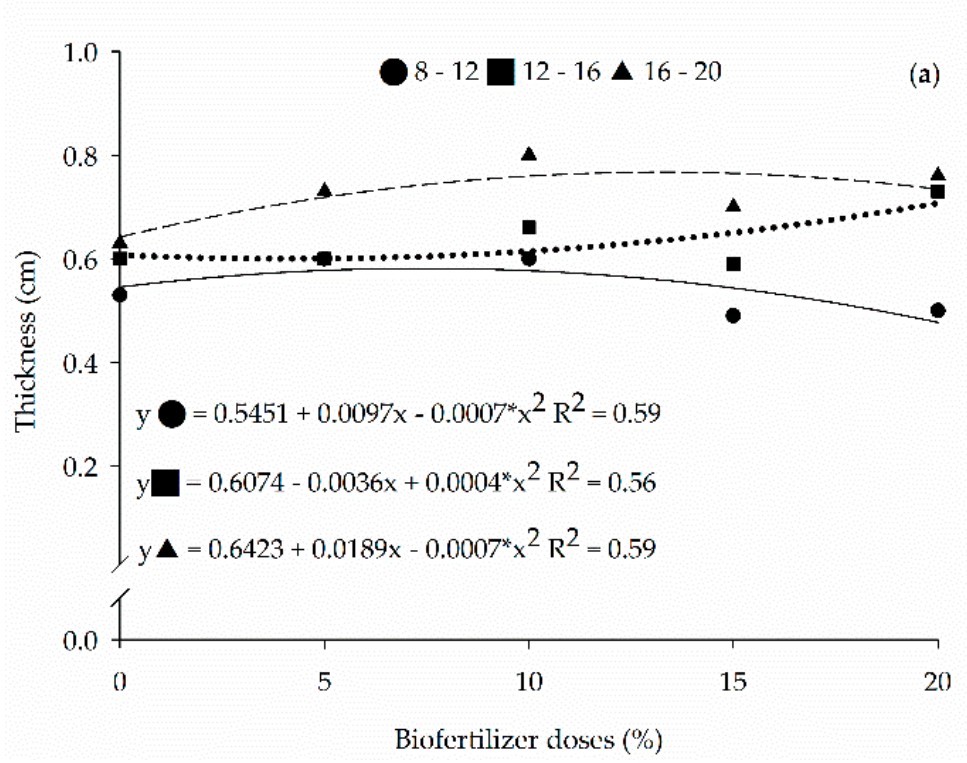

**Figure 2.** *Cont*.

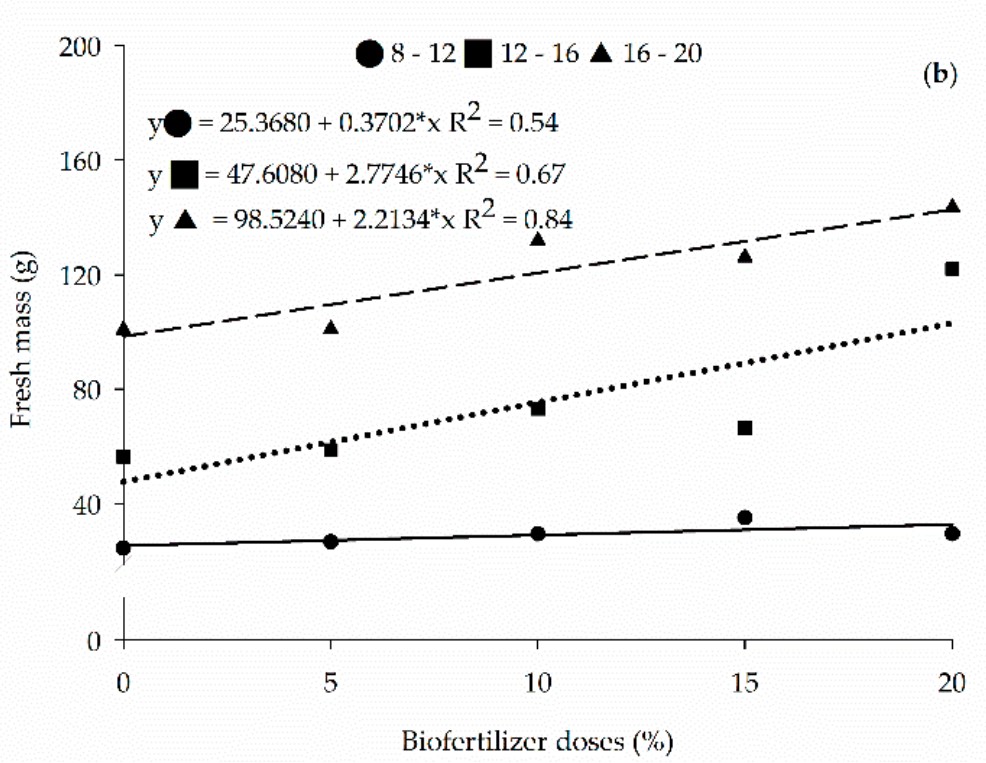

**Figure 2.** (**a**) Thickness in centimeters (cm) and (**b**) fresh mass in grams (g) for cladodes of *Opuntia stricta* (Haw.) Haw. cultivated with biofertilizer. * indicate significant differences at 5% probability.

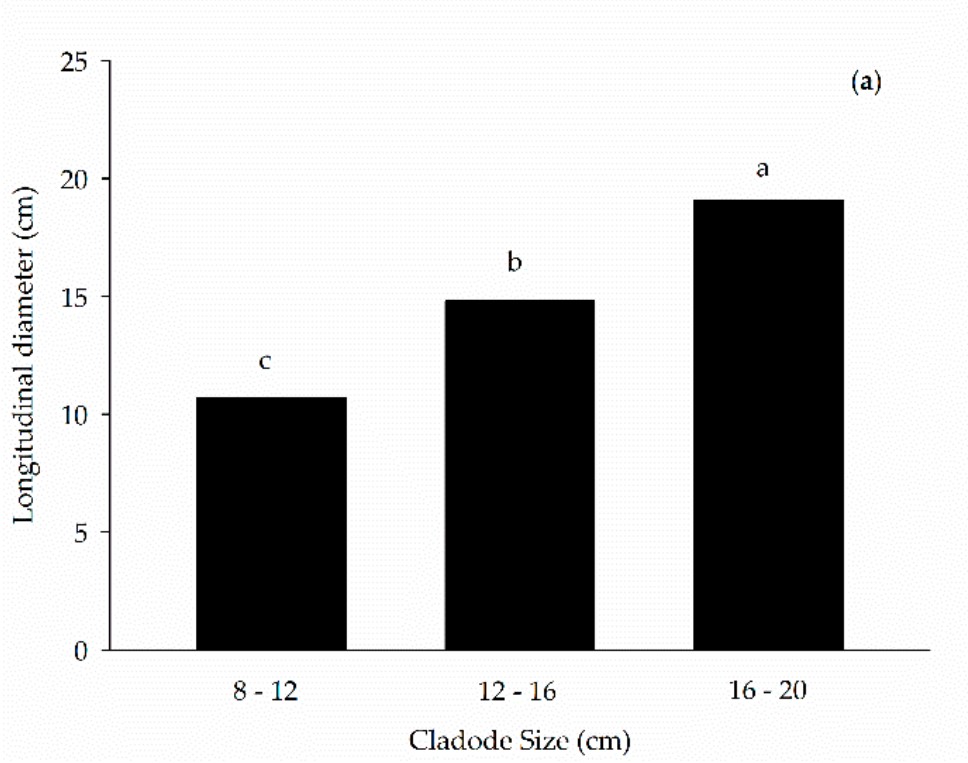

**Figure 3.** *Cont.*

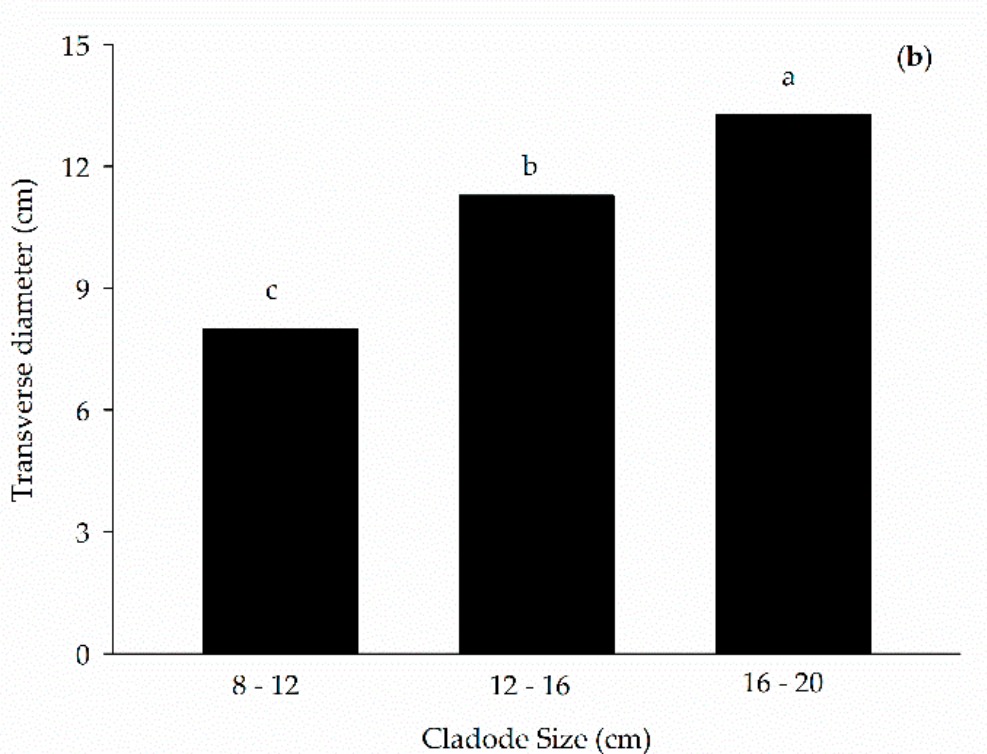

**Figure 3.** (**a**) Longitudinal diameter and (**b**) transverse diameter of *Opuntia stricta* (Haw.) Haw. cactus cladodes in centimeters (cm). Means followed by same letter are not statistically different using Tukey test at 5% (*p* > 0.05).

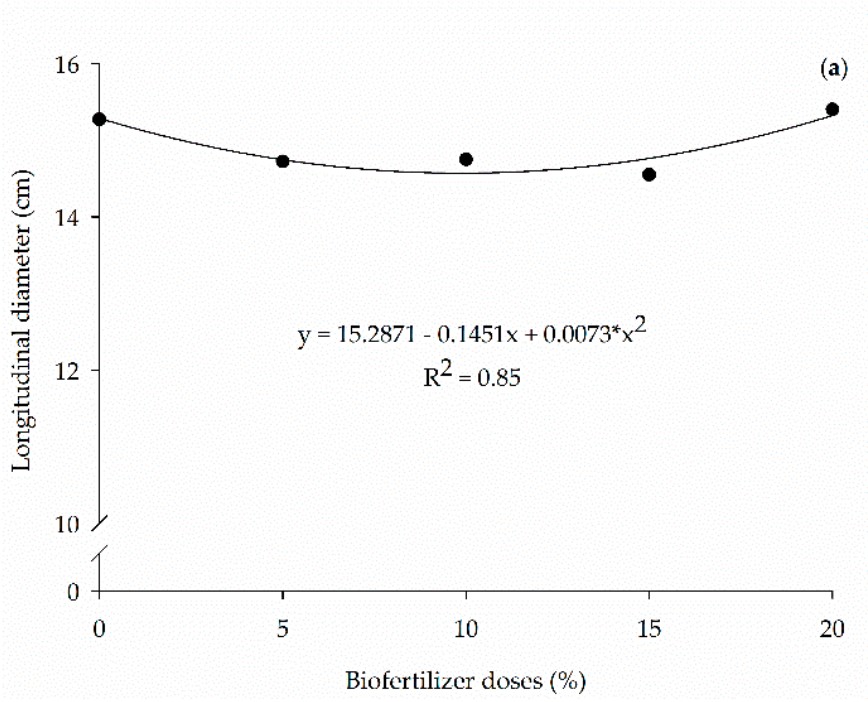

**Figure 4.** *Cont.*

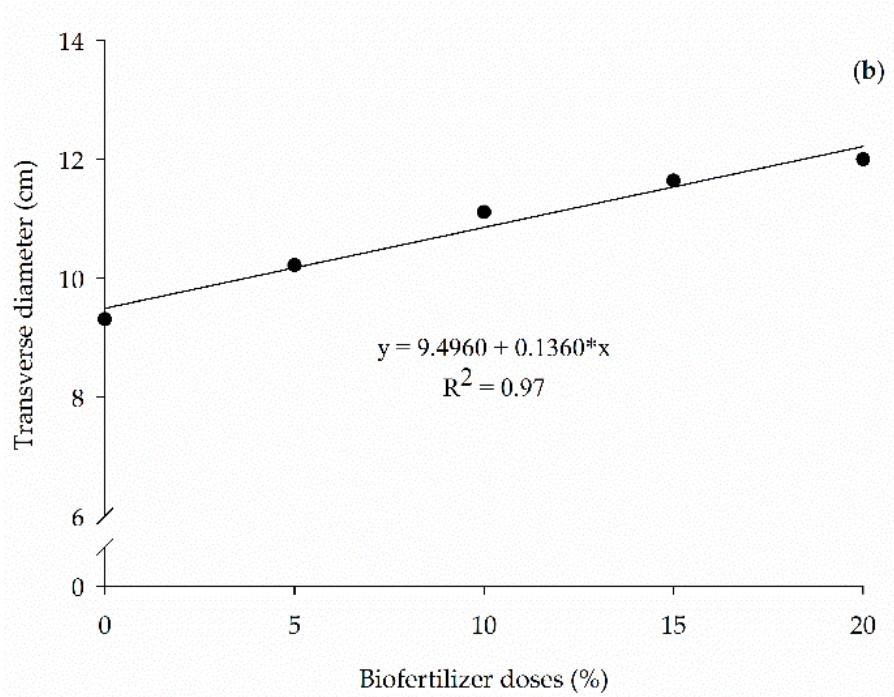

**Figure 4.** (**a**) Longitudinal diameter and (**b**) transverse diameter of *Opuntia stricta* (Haw.) Haw. cactus cladodes in centimeters (cm) with increasing biofertilizer doses (0% to 20%). * indicate significant differences at 5% probability.

According to the analysis of variance for the physical-chemical parameters, there was an interaction between the doses of biofertilizer and the cladode length for pH, H+ ions, soluble solids (SS), soluble solids/titratable acidity ratio (SS/TA), and ascorbic acid (AA) (Table 4). The pH of palm cladodes increases as a function of the length and dose of the biofertilizer. Observing the pH, the greatest values were 4.38, 4.33, and 4.05 for the cladode lengths of 8–12, 12–16, and 16–20 cm, respectively, reached at the biofertilizer doses of 10, 15, and 20%. It can be seen in Figure 5a. For the concentration of hydrogen ($H^+$) ions, the cladodes 12–16 cm long were greater in comparison with the others, with the greatest value at the dose of 0% biofertilizer, with an average of 227.43 µM, followed by cactus cladodes with a length of 8–12 cm, with an average of 217.19 µM at the same dose of biofertilizer. As for cladodes that were 16–20 cm long, the greatest value was at the dose of 5% biofertilizer, with an average of 157.33 µM. Thus, pH was inversely proportional to the increase in biofertilizer doses, indicating that increasing biofertilizer application increases the concentration of $H^+$ ions (Figure 5b).

**Table 4.** Summary of analysis of variance for pH, H+ ions, soluble solids (SS), titratable acidity (AT), soluble solid ratio and titratable acidity (SS/TA), and ascorbic acid (AA) of palm cladodes *Opuntia stricta* (Haw.) Haw. fertilized with different doses of biofertilizer.

| Variation Sources | Degrees of Freedom | Medium Squares | | | | | |
|---|---|---|---|---|---|---|---|
| | | pH | $H^+$ | AT | SS | SS/TA | AA |
| Doses (D) | 4 | 0.10 ** | 4951.2 ** | 0.0099 ns | 1.02 ** | 0.98 ** | 28.25 ** |
| Length (C) | 2 | 0.28 ** | 22,189.3 ** | 0.054 ns | 7.65 ** | 6.65 ** | 17.01 ** |
| D × C | 8 | 0.11 ** | 5776.7 ** | 0.058 ns | 1.18 ** | 1.22 ** | 9.36 ** |
| Residue (error) | 28 | 0.0012 | 125.79 | 0.035 | 0.086 | 0.099 | 0.62 |
| Average | | 2.93 | 129.98 | 1.73 | 4.18 | 2.45 | 19.47 |
| Coefficient of Variation (%) | | 0.59 | 8.63 | 10.1 | 7.02 | 12.84 | 4.06 |

The symbol ns denotes differences between factors are not significant, while ** indicate significant differences at 1% probability, respectively, using the F-test.

The soluble solid content was greater in the cactus cladodes of 12–16 cm (6.69%) at the dose of 15% in comparison with the cladodes of 8–12 (5.53%) and 16–20 cm (4.13%) for the doses of 10 and 15%, respectively (Figure 5c). The SS/TA ratio of the cladodes of 12–16 cm at the biofertilizer dose of 15% (4.42) surpassed those of the cladodes of 8–12 and 16–20 cm, with SS/TA values of 2.90 and 2.57 at the biofertilizer doses of 0 and 15%, respectively (Figure 5d). The content of ascorbic acid was greater for cactus cladode lengths of 8–12 cm and 12–16 cm (22.36 mg/100 g) at biofertilizer doses of 10 and 5% when compared with the cladodes 16–20 cm long (20.30 mg/100 g) at the biofertilizer dose of 20% (Figure 5e).

Also significant were the effects of the interaction between the doses of biofertilizer and the length of the cladode on the contents of chlorophyll, carotenoids, phenolic compounds, and total sugars (Table 5). For the chlorophyll a and total contents, there was an increase in values depending on the size of the cladodes, with the highest values being observed for (0.69, 0.64, and 0.75 mg/100 g) and (1238.2, 1249.9, and 1446.0 mg/100 g) for cladode lengths of 8–12, 12–16, and 16–20 cm, respectively, both for the 0% biofertilizer dose (Figure 6a,b). The chlorophyll *b* contents were 0.53, 0.70, and 0.69 mg/100 g for the cladodes of 8–12, 12–16, and 16–20 cm at the doses of 0, 15, and 0% (Figure 6c). The greater carotenoid contents were 78.81, 80.81, and 96.74 µg/100 g in the cladode lengths of 8–12, 12–16, and 16–20 cm at the 0% and 20% doses of biofertilizer, respectively (Figure 6d).

The greatest total sugar contents of 0.50%, 0.23%, and 0.25% were at biofertilizer doses of 0%, 5%, and 15% and at cactus cladode lengths of 8–12 cm, 12–16 cm, and 16–20 cm, respectively. Therefore, the cladode length of 8–12 cm generated increased sugar contents (Figure 6e). The phenolic compound contents presented an increase at the biofertilizer level of 20% for both length groups, with values of 8.53, 8.94, and 9.44 mg/100 g in the cladodes with lengths of 8 to 12, 12 to 16, and 16 to 20 cm for each unitary increase in the biofertilizer doses, respectively (Figure 6f).

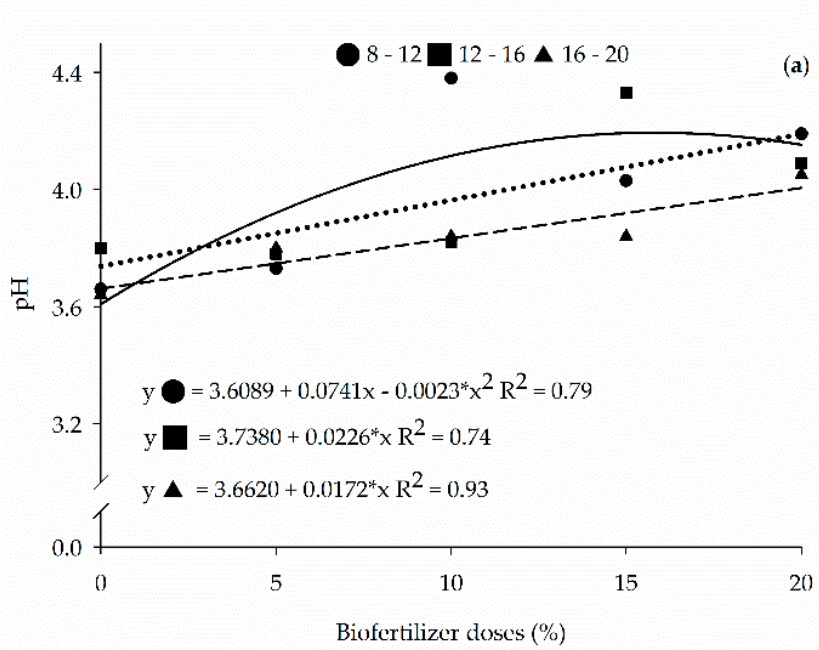

**Figure 5.** *Cont.*

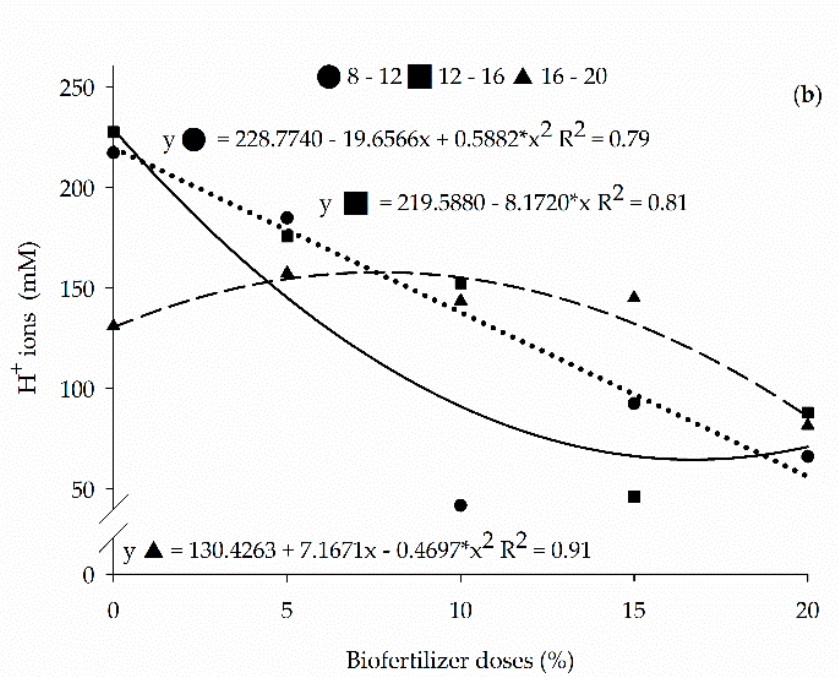

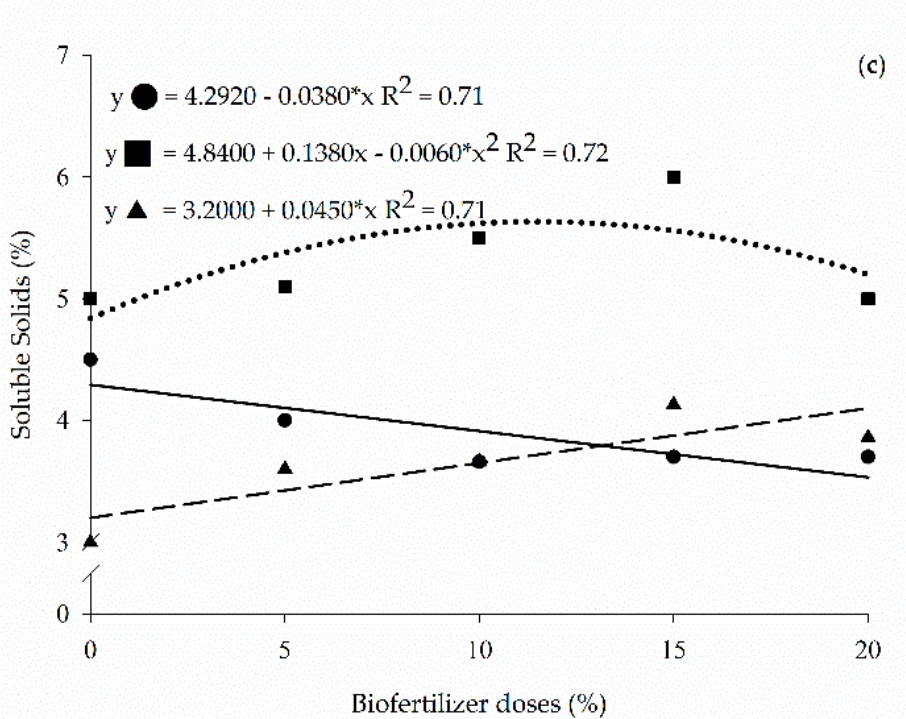

**Figure 5.** *Cont.*

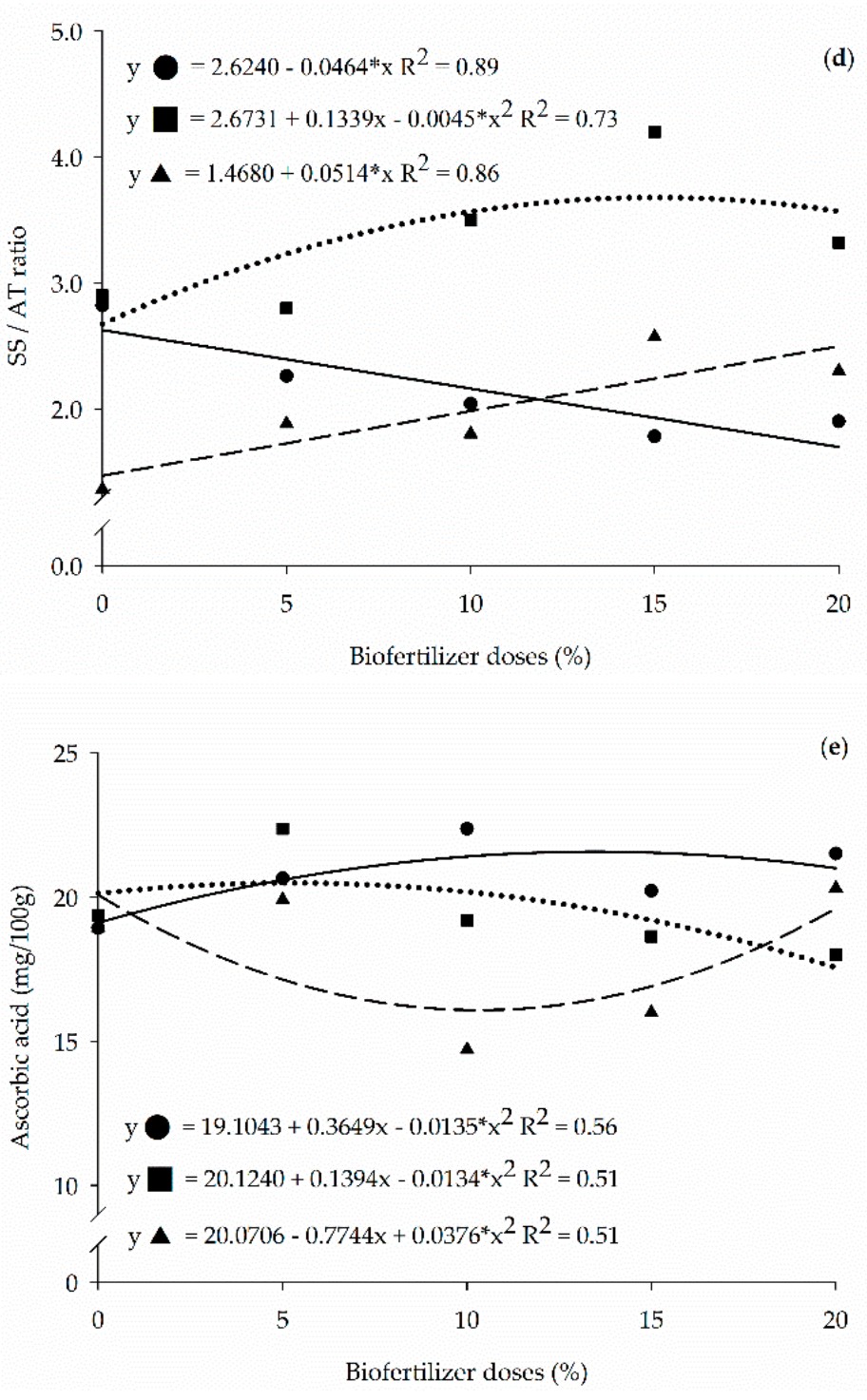

**Figure 5.** (**a**) pH, (**b**) H + ions, (**c**) soluble solids (%), (**d**) soluble solids/titratable acidity ratio (SS/TA), and (**e**) ascorbic acid (milligrams per 100 g) of *Opuntia stricta* (Haw.) Haw. cactus cladodes of different lengths and applied doses (0% to 20%) biofertilizer. * indicate significant differences at 5% probability.

**Table 5.** Summary of analysis of variance of *Opuntia stricta* (Haw.) Haw. cactus cladode chlorophyll *a* (Clo *a*), chlorophyll *b* (Clo *b*), total chlorophyll (total clo), carotenoids (Car), phenolic compounds (Phnl.comp.), and total sugars for different cladode lengths and biofertilizer doses.

| Variation Sources | Degrees of Freedom | Medium Squares | | | | | |
|---|---|---|---|---|---|---|---|
| | | Clo *a* | Clo *b* | Clo Total | Car | Phnl.comp. | Total Sugars |
| Doses (D) | 4 | 0.014 ** | 0.021 ** | 18,863.1 ** | 2797.2 ** | 1.85 ** | 0.016 ** |
| Length (C) | 2 | 0.10 ** | 0.058 ** | 286,522.4 ** | 2357.1 ** | 2.65 ** | 0.013 ** |
| D × C | 8 | 0.012 ** | 0.027 ** | 67,914.7 ** | 258.27 ** | 0.60 ** | 0.026 ** |
| Residue (error) | 28 | 0.00015 | 0.00020 | 367.45 | 6.11 | 0.0071 | 0.000010 |
| Average | | 0.53 | 0.52 | 1057.5 | 60.1 | 8.10 | 0.21 |
| Coefficient of Variation (%) | | 2.33 | 2.76 | 1.81 | 4.12 | 1.04 | 1.48 |

The symbol ** indicates significance at the 1% probability using the F-test.

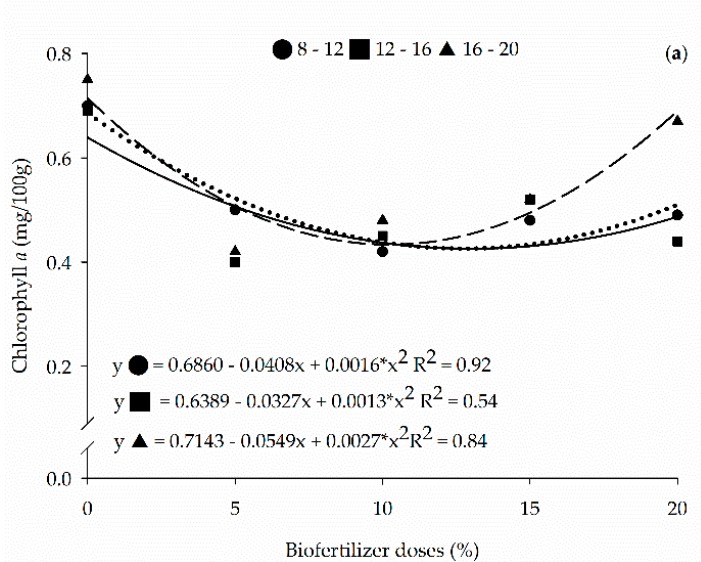

$$y \bullet = 0.6860 - 0.0408x + 0.0016 \ast x^2 \quad R^2 = 0.92$$
$$y \blacksquare = 0.6389 - 0.0327x + 0.0013 \ast x^2 \quad R^2 = 0.54$$
$$y \blacktriangle = 0.7143 - 0.0549x + 0.0027 \ast x^2 \quad R^2 = 0.84$$

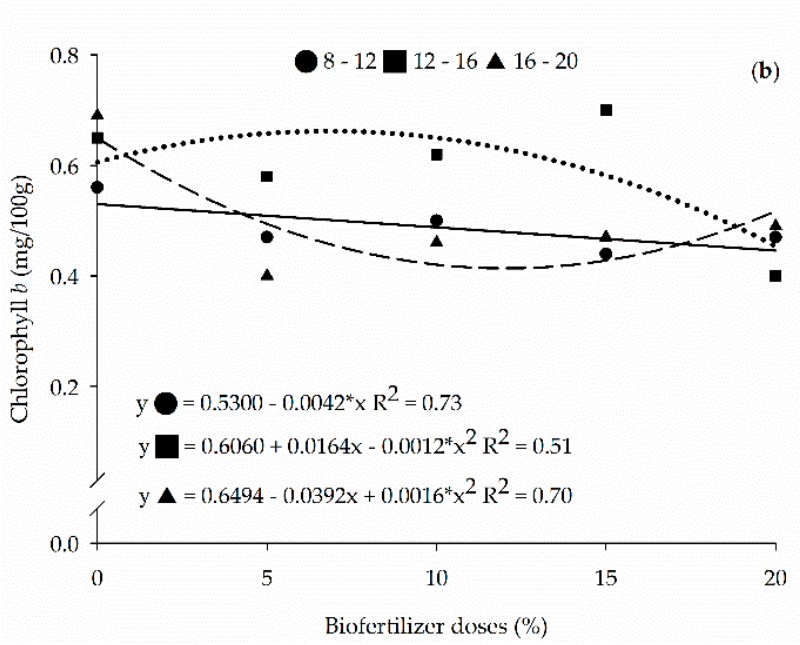

$$y \bullet = 0.5300 - 0.0042 \ast x \quad R^2 = 0.73$$
$$y \blacksquare = 0.6060 + 0.0164x - 0.0012 \ast x^2 \quad R^2 = 0.51$$
$$y \blacktriangle = 0.6494 - 0.0392x + 0.0016 \ast x^2 \quad R^2 = 0.70$$

**Figure 6.** *Cont.*

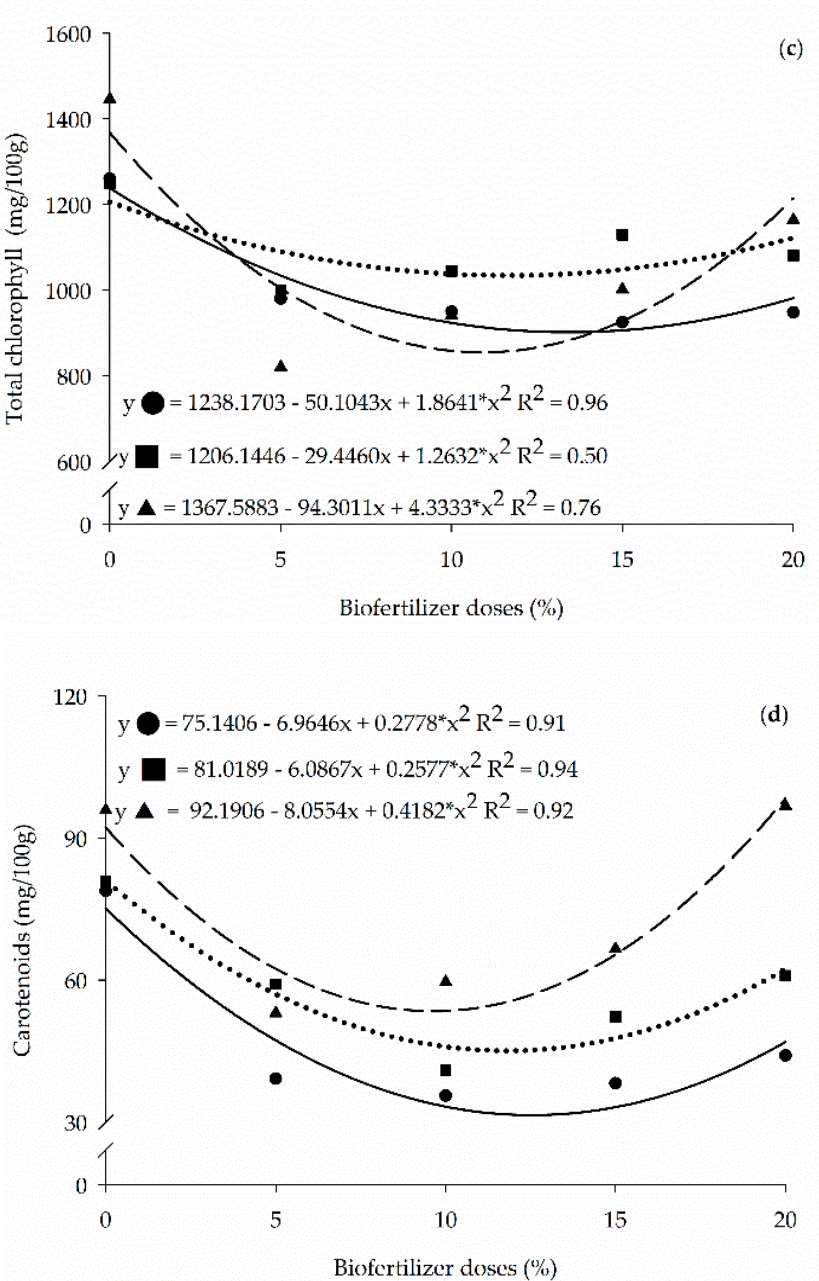

**Figure 6.** *Cont.*

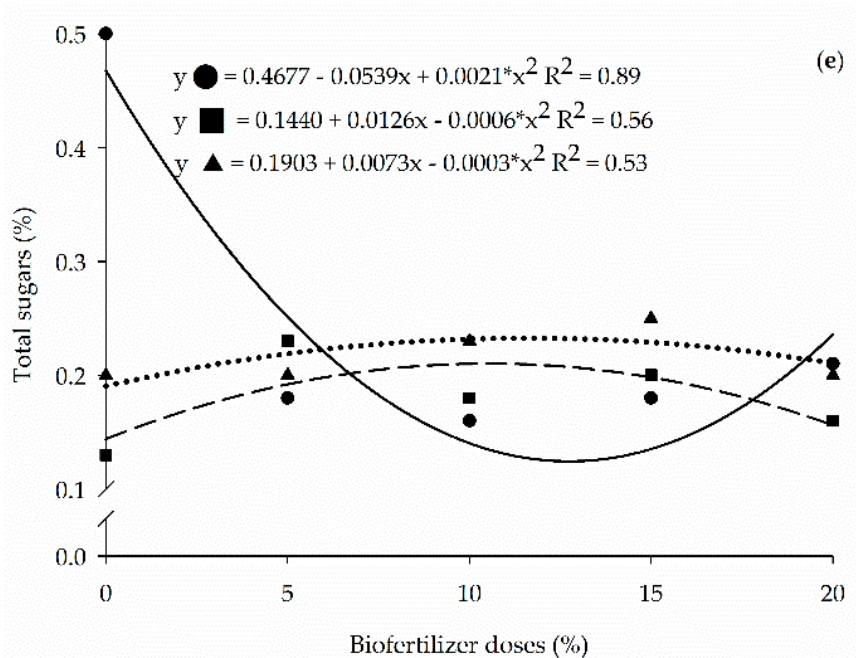

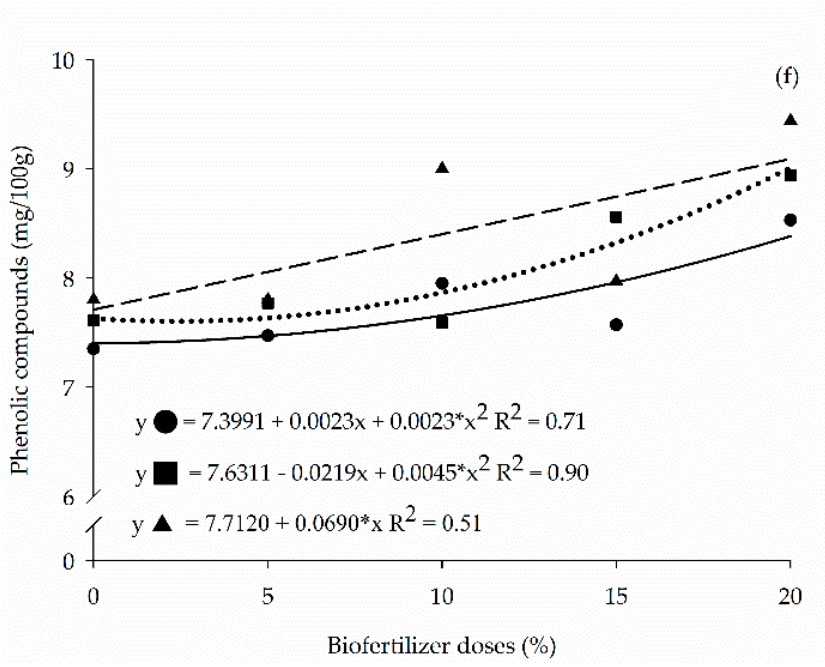

**Figure 6.** Indices of (**a**) chlorophyll a, (**b**) chlorophyll b, (**c**) total chlorophyll, (**d**) carotenoids, (**e**) total sugars, and (**f**) phenolic compounds of *Opuntia stricta* (Haw.) Haw. cactus cladodes of different lengths and applied doses of biofertilizer. Except for total sugars (%), all indices measured in milligrams (mg) per 100 g (g) of cactus cladode. * indicate significant differences at 5% probability.

As shown in Figure 7, young cactus cladodes with biofertilizer had great similarity in their respiratory rates. The cactus cladode size of 4–8 cm, for both treatments, presented higher respiratory rate values of 344.10 mg of $CO_2$/kg/h for those grown without biofertilizer and 278.85 mg of $CO_2$/kg/h for those grown with biofertilizer. The respiratory rate of the cladodes 4 to 8 cm long grown without biofertilizer was superior to the other sizes with an average of 344.10 milligrams (mg) of $CO_2$/kg/h, which was superior to the cactus cladodes grown with biofertilizer (278.85 mg $CO_2$/kg/h) in the same size group.

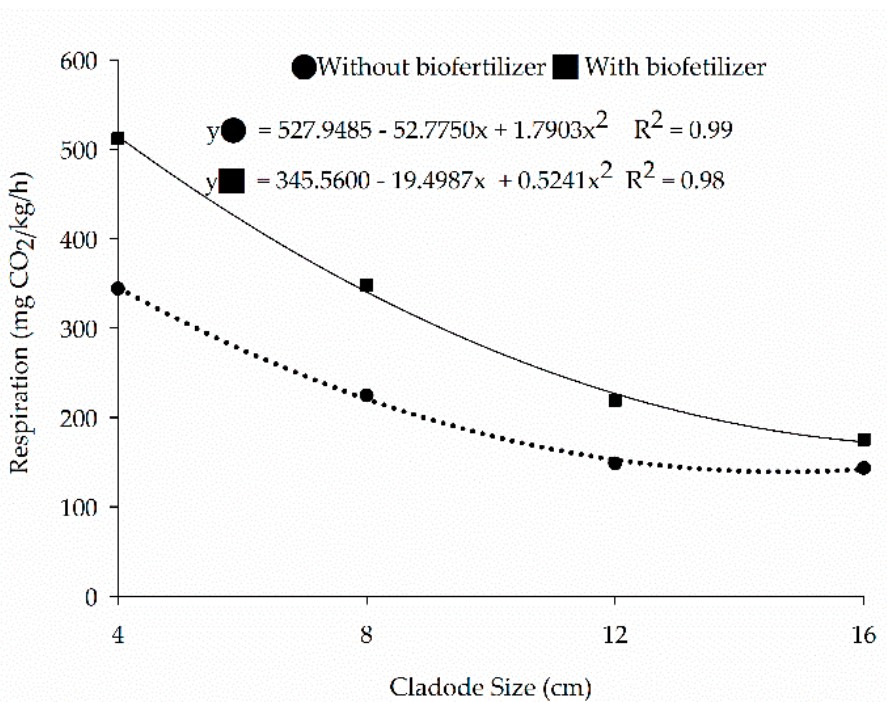

**Figure 7.** Respiratory rates of *Opuntia stricta* (Haw.) Haw. cactus cladodes of different lengths, both with and without biofertilizer, measured in milligrams (mg) of $CO_2$ per kilogram (kg) of cactus cladode per hour.

## 4. Discussion

### 4.1. Underlying Drivers of Cactus Responses to Biofertilization

Foliar fertilization via biofertilizer improves the physical attributes of palm cladodes, enabling greater productivity and yield. These results suggest that liquid biofertilizers increase cactus development, providing a better nutritional profile based on the availability of nutrients. Biofertilization of *Opuntia stricta* (Haw.) Haw. favored an increase in physical attributes and biomass production. A greater supply of organic matter in the soil enables an increased nutritional and physiological balance, consequently favoring the production and accumulation of photoassimilates [24]. The provision of organic matter to the soil through biofertilizers promotes improvements to the plant's physical attributes. Since this material is in a liquid state, which is easily absorbed, it generates better conditions for the plants' development, improving their physical and nutritional performance. These results are similar to those obtained by [1] when working with organic fertilization, where increases in physical attributes were observed for plants that were fertilized with cattle manure, with values ranging from 10.9 to 18.3 cm. The transversal and longitudinal diameters were superior for cladodes grown at a biofertilization level of 20% at 12.20 cm and 15.40 cm, respectively.

The greatest longitudinal and transversal diameters of the cladodes with lengths of 16–20 cm occurred because they were in a development stage that was superior to the others, since a larger diameter is an indicator of cactus cladodes in more advanced development stages. Prior research [25], working with different length groups of giant and round *Opuntia* species, also noticed that the larger the cladode size, the greater the values of cladode diameters, and with 74.02% and 96.01% greater yields, while comparing the results from bigger and smaller cladodes, respectively.

As the biofertilizer dose increased, the pH value tended to increase, regardless of the cladode lengths, because the pH of the biofertilizer used in the research was alkaline. Another explanation for this difference in the cladodes' pH values can be related to the Crassulacean acid photosynthetic metabolism (CAM) pathway, which has the tendency to present acidity variation [26]. The variation in the $H^+$ ion concentration in cladodes has

been reported in other research, such as [27], which observed variation between 129.7 and 344.4 μM in cladodes of the erect prickly pear species of different sizes. This variation can be driven by the cladodes' development [26].

It has been reported that the increase in the cladodes' length results in greater soluble solid contents. The high soluble solid content in the cladodes 12–16 cm long can be related to environmental conditions, directly affecting the plants' metabolism during harvest, as reported by [28]. The greatest soluble solid (SS) to titratable acidity (TA) or SS/TA ratio in the cladodes 12–16 cm long at 15% biofertilizer is an indication that these cladodes present pleasant characteristics to the palate, since the greater the SS/TA ratio, the greater the concentration of soluble solids, and the more pleasant the taste, as reported by [27].

These values are superior to those reported by [1], where there was a reduction in ascorbic acid as the organic fertilization doses were increased, presenting maximum values of 7.54 milligrams (mg) per 100 g (g). Prior research [5], while working with different cactus species, reported values of 5.7, 4.3, and 1.6 mg of ascorbic acid/100 g of cactus in the cultivars *Opuntia strica*, *Opuntia fícus-indica*, and *Nopalea cochenillifera*, respectively. Similar results were reported by [29], where increasing amounts of biofertilizer were applied to beets, which also resulted in an increase in ascorbic acid content.

For both types of chlorophyll, there are variations in the chlorophyll contents with increased biofertilizer doses. These variations may have been caused because both chlorophylls, especially chlorophyll *a*, and carotenoids have the tendency to be inversely proportional to biofertilizer dose. Nonetheless, the species seems to adjust itself to the increased supply of nutrients, where there is a tendency for elevated chlorophyll contents associated with biofertilizer doses greater than 10%. The greatest chlorophyll values were associated with a cactus cladode length of 12–16 cm at the biofertilizer dose of 15%. The successive application of organic fertilizer over a longer period can increase the nutrient source in the soil, promoting the supply of nitrogen (N) and magnesium (Mg), in addition to other nutrients that take part in the synthesis of this pigment. These chlorophyll *b* variations can be related to several factors, such as the species and cultivar, as well as environmental conditions, such as luminosity and temperature, which can promote its degradation, as reported by [1] for *Opuntia stricta*.

We reported a greater carotenoid content for cactus cladodes with lengths of 16–20 cm at the biofertilizer level of 20%. The more developed cladodes had greater carotenoid contents because they already had a darker color in comparison with the younger cladodes. This is also observed in vegetables with more intense colors and more advanced development. These pigments' evaluation is important because they have antioxidant and anticancer effects [28].

The increase in the size of the cladodes induces a greater production of phenolic compounds. As the cladode expands and grows older, there is greater accumulation and variation of photoassimilates and reserves that are constantly changed into other compounds, such as phenolics. Farias [30] reported similar behavior in 'Giant' (*Opuntia ficus indica* L. Miller) and 'Round' (*Opuntia* spp.) cactus cladodes.

We observed an inverse relationship between cladode maturity and total sugar content. During their development, cactus cladodes reach greater physiological maturity, which reduces total sugar content. This fact was verified for giant and round cactus cladodes, where sugar contents ranging from 0.59% to 0.72% and from 0.52% to 0.72% were obtained, respectively [30]. These low levels may be related to the presence of other substances dissolved in the aqueous medium of the plant, such as organic acids, pectins, mucilages, and phenolics.

Increasing biofertilizer doses promote greater cactus cladode quality, regardless of cladode length. This effect is likely driven by the greater availability of nutrients from biofertilization, which results in more nutrients being absorbed by the plants and resulting in more plant development. This happens because these compounds have nitrogen in their composition, which is accumulated to act especially in the plant's defense [31]. Therefore,

the provision of organic matter to the soil via biofertilizer promotes improvements to the soil's characteristics, consequently improving plant development [32].

Regardless of the production method, respiration tends to decrease as the cladodes' maturation stage advances due to greater energy consumption, just as the respiratory rate increases in younger cladodes when they are removed from the plant. The cladodes grown with biofertilizer presented lower respiratory rates. This condition can be related to an increased supply of nutrients from biofertilization, which enables the construction of several molecules that assist both in the protection of and in the transportation of nutrients in the plants. When the biofertilizer is applied foliarly, the absorption of nutrients is stimulated, especially nitrogen, which has a role in the assimilation of countless amino acids, which are then incorporated into proteins and nucleic acids, which mold chloroplasts, mitochondria, and other structures that hold most of the biochemical reactions [33].

According to [34], plant respiration involves the oxidative decomposition of complex substances present in the cells (such as starch, sugars, and organic acids) into simple molecules ($CO_2$ and $H_2O$) for energy production. Generally, the respiratory rate is directly linked to the deterioration rate of a harvested product, and the temperature to which it is exposed directly affects the respiration. In other words, the increase in temperature raises the respiratory rate, therefore decreasing its post-harvest life. The results showcased in our research confirmed our hypothesis that biofertilizers increase the physical and physical-chemical qualities of cladodes. Nonetheless, bigger cladodes are more responsive to the use of biofertilizer.

*4.2. Human Nutrition*

*Opuntia* species have had a long history of use for both human food and animal feed in the Americas. These cacti have been used both pre- and post-colonization by both indigenous peoples and colonial settlers, particularly in Mexico [35]. The use of such cactus species has been and can continue to be relevant in adapting to drought given the limits and challenges of fighting against drought by increasing water sources (e.g., reservoirs, wells) [36].

*Opuntia* is a genus that plays a strategic role in agriculture in regions with arid and semi-arid climates. In addition to the potential for adapting to edaphoclimatic conditions, *Opuntia* spp. have nutraceutical compounds that promote improvements in human nutrition and health, establishing food security, in addition to being a species with applications in the food industry [37,38]. According to [38], this cactus genus has high levels of phosphorus (62.2 mg/100 g), sodium (9.1 mg/100 g), calcium (1.18 to 1.28 mg/100 g), iron (3.0 to 3.8 mg/100 g), potassium (45–47 mg/100 g), and flavonoids (196.7 mg/100 g). This demonstrates this cactus' social and cultural importance in rural communities, especially in the Brazilian Northeast. In our study, the increase in phenolic compounds with the application of the biofertilizer was desirable, as it increased the benefits of inserting *Opuntia* spp. in human food without worrying about toxicity [39].

In addition, the benefit of consuming *Opuntia* spp. is also due to the cladodes, pulp, fruits, seeds, and bark having high levels of antioxidants. This type of cactus is also anti-diabetic, anti-bacterial, anti-viral, anti-inflammatory, and analgesic, and it is used in neopathy and folk medicine [39,40]. According to [41], *Opuntia* spp. can be used in the preparation of cosmetics and dyes, adding value and becoming a raw material of economic interest. Although *Opuntia* is a genus with diverse nutraceutical characteristics, few people adopt its use in human food in Brazil, although this can vary according to cactus distribution, cultural aspects, and even differences in family financial backgrounds. Thus, our study describing its physicochemical characteristics makes its use for human food and supplementation feasible. According to [42], these cacti have a lot of potential to combat hunger. In addition, products made with *Opuntia* have good sensory acceptance [43] and can be a viable option to encourage sustainable agricultural development in Brazil.

### 4.3. Management and Ideal Size of Cactus Cladodes for Human Consumption

Generally in Brazil, cactus is used for animal feed, making its management carried out with low intensity and few technologies being used by producers. This is true not only during planting but also later during the growth and development phases of the crop. Normally, cactus producers in the northeast region of Brazil make use of organic fertilizers such as bovine, goat, and poultry manure, which are added in furrows or holes during planting. The use of organic amendments in agricultural systems for cacti has been widespread, given their important economic and environmental contributions [44].

From an economic point of view, cactus producers can realize increased income due to the decrease in the use of industrial fertilizers. This makes cactus cultivation possible for less capitalized farmers, which can generate promising results from such mineral fertilization [44].

Other studies verify the benefits of using organic fertilizers during cactus production. For example, researchers have found that cacti respond well to organic fertilization since soils in semi-arid regions tend to have low organic matter content. Therefore, organic fertilization with livestock manure can increase organic matter in these soils and be essential to crop production [45]. It is recommended to fertilize cacti with bovine and goat manure at a rate of 10 to 30 metric tons /hectare during planting and then every two years during the beginning of the rainy season [46]. The presence of animal manure, mainly from cattle, on most agricultural properties can help maintain soil fertility in Brazil's semi-arid regions [47].

Few studies have focused on the cultural management of *Opuntia* spp. cacti in Brazil. Most research has focused on the use of cactus cladodes in human food as well as the manufacture of differentiated, value-added products [30]. Research is needed to determine the ideal size of cactus cladodes for human consumption.

In general, in Brazilian cuisine, there is no consumption of cactus cladodes. In some states of Brazil, there is a small trade in such cladodes, mainly at small urban fairs. Thus, this work is a first step in characterizing the post-harvest quality and size of *Opuntia* spp. cladodes intended for human consumption using organic fertilizer. Our research results suggest that cactus production is a viable commercial alternative for small- and medium-sized cactus producers in Brazil's semi-arid regions. It is important for cactus cladodes to be balanced in terms of palatability, measured as the soluble solid to titratable acidity ratio (SS/TA), for the more mature developmental stages of the crop.

Likewise, it was observed for the 'Redonda' and 'Gigante' cacti cultivars that yield, physical attributes, and bioactive compounds of the cactus buds increased with developmental stage advancement up to 20 cm, which is the ideal for consumption [30]. This same study also reported that yield varied according to the size of the cactus cladodes and the number of thorns [30]. Cactus yield for food can play an important role when selecting the best-sized shoots to optimize propagation for both consumption and agro-industrial processing.

In recent years in Brazil, some authors have determined that the cactus cladode size of 15 to 20 cm is ideal for human consumption. Cladodes should be harvested 30 to 60 days after sprouting and weigh 80 to 120 g. They must also be tender, young, fine, fresh in appearance, turgid, and bright green in color so that they can also be used in various culinary recipes from around the world [30]. In Mexico, researchers have determined that the ideal size of cactus cladodes for human food is 10 to 12 cm in length and that the cladodes need to be tender [48]. Another study concluded that ideal cladodes were 20 cm long and weighed 90 to 100 g [49]. Cladodes to be exported should be between 17 and 21 cm in length, and those selected for national consumption should be between 21 and 25 cm [50], while [51,52] state that cladodes are ready for consumption when they are between 7 and 30 cm long.

Mexico is one of the origin centers of *Opuntia* spp., where consumption of cladodes is ancestral [53]. Cladodes are consumed as vegetables. Harvested area has more than doubled from 5269 to 12,105 ha between 1990 and 2012 [50]. Cactus plants, due to their adaptation to moisture deficits and semi-desert and desert climates, are a food source with

great potential for the development of plantations [54]. Mexico is the main producer of cactus cladodes in the world, accounting for 74% of global production, and is the largest consumer of both fresh and processed cactus. However, other markets, such as those in the United States and Canada, present a growing opportunity for export [50]. Due to their nutraceutical characteristics, cactus cladodes have shown great interest in the world market [53], as they are beneficial for various treatments [50]. Furthermore, they contain vitamin C, minerals, as well as soluble and insoluble fiber [50,53].

For the commercial use of cactus cladodes in Mexico, producers already use commercial size standards. However, it is important to follow similar commercial standards for export and national trade. These standards do not yet exist in Brazil since consumption of cactus cladodes in Brazil is still regional and limited. Further research and incentives from governments and public entities are needed in order to expand the production and consumption of cactus as a vegetable. Brazil has great potential for developing cactus as an export crop since it is tied with Tunisia in global production (600,000 hectares for each country), with Mexico at third with 230,000 hectares of cultivated cactus [55]. This holds great importance not only for Mexico but also for other countries and markets around the world.

## 5. Conclusions

The *Opuntia stricta* cladodes with a length of 16 to 20 cm present greater physical and physical-chemical attributes when submitted to a range of biofertilizer doses. The biofertilizer dose of 20% promotes an increase in the contents of soluble solids, phenolic compounds, and total sugars. Nevertheless, it did not affect the content of pigments. The cladodes (4–8 cm long) grown with and without biofertilizers presented greater respiratory rates. The use of biofertilizer is a promising practice for the cultivation of *Opuntia stricta* in agroecological or organic systems. The physical-chemical characteristics presented in our study provided information that enables the use of this cactus in human food and other alternatives for economic income.

**Author Contributions:** Conceptualization, M.S.S.; methodology, M.S.S. and J.S.N.; software and validation, C.C.S.; formal analysis, F.B.C.; investigation, D.C.A.; resources, A.K.H. and W.M.S.; data curation, M.S.S.; writing—original draft preparation, writing—review and editing, A.K.H., C.C.S., F.B.C., J.A.L. and M.S.S.; visualization, supervision, U.S.P. and F.A.L.G.; project administration, S.P.Q.S.; funding acquisition, M.S.S. All authors have read and agreed to the published version of the manuscript.

**Funding:** This research was funded by CNPq-Conselho Nacional de Desenvolvimento Científico e Tecnológico, grant number 456518/2014-2.

**Institutional Review Board Statement:** Not applicable.

**Informed Consent Statement:** Not applicable.

**Data Availability Statement:** Not applicable.

**Acknowledgments:** I would like to thank the CAPES (Coordenação de Aperfeiçoamento de Pessoal de Nível Superior) by financial assistance (001), Food Science, Technology and Engineering Research Group (GPCTEA), AgriSciences, the collaborators of Revista Sustentabilidade (MDPI), and all the partners who contributed to the planning, execution, writing, and publication of the work. We also thank two anonymous reviewers whose comments and edits improved the quality of our work.

**Conflicts of Interest:** The authors declare no conflict of interest.

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
