# Peer review of "Organic Fertilization with Biofertilizer Alters the Physical and Chemical Characteristics of Young Cladodes of Opuntia stricta (Haw.) Haw."

_sustainability, doi:10.3390/su15043841_

Round 1

Reviewer 1 Report

Dear authors: 

After reading the manuscript “Physical and chemical characteristics of young cladodes of Opuntia stricta (Haw.) Haw. with doses of bio-fertilizer”, the comments and suggestions are the following.

1. Change the title. The authors indicate that they worked with bio-fertilizers but have not mentioned what kind of microorganisms were developed with the fermentation of bovine manure. It is more appropriate to use organic fertilization.

2. In Mexico, organic fertilization and irrigation are widely used in the production of cladiolus for human consumption, preferably small in size. However, for livestock consumption, they are not fertilized, and cladioles are allowed to grow up to 20 cm. These are more fibrous and may be of higher quality in terms of sugars, but they are not palatable.

3. In the literature review, the authors concentrated on research carried out in their country but did not carry out a much more detailed analysis of cladiolus and the various variables to be studied that have been carried out in other countries. Furthermore, they need to indicate the management of Opuntia in their country, the problems they have, production costs, and even acceptance by the population in terms of size. In such a way that they do not justify the study or the problem they want to solve (is there a problem with the size of the cladiolus, fertilization, production costs they want to lower, etc.), that would allow them to put forward hypotheses more related to the research. 

4. The hypothesis is that the "increase of the biofertilizer's availability is directly proportional to the physical and physical-chemical qualities of the Opuntia stricta regardless of the growth stage evaluated. However, as the research question or the contribution to the generation of knowledge on Opuntia production needs to be established, the hypothesis is not a tentative answer.

5. Objectives: to evaluate the physical and physical-chemical qualities of different lengths of Opuntia stricta cladodes grown with different doses of biofertilizer. They further add that the objectives are for:

a) low-cost organic matter (e.g., biofertilizer) with a direct effect on the physicochemical quality of the cladodes for fresh consumption as an unconventional vegetable.

b) Cultivation techniques that can improve cladode and fruit quality are practices which are decisive in the marketing of crop yield.

c) To improve the market value of the crop and the economic income of farmers and rural populations in arid and semi-arid regions in Brazil and elsewhere in the world.

The authors did not assess, nor did they indicate, the current costs of cladiolus production and how much these costs would be reduced by applying organic fertilizers or bio fertilizers. Furthermore, they do not indicate or survey consumer opinion on cladiolus size, let alone mention how to improve the market value of the crop and the economic income of farmers and rural populations.

Materials and Methods

It is recommended to separate into subheadings:

Site location and Laboratory analysis. The laboratory analyses recommend subdividing Physical-chemical analyses, Nutritional characteristics, respiratory rate, and Statistical analyses).

It is also recommended to indicate the manuals or authors from which the procedures of each section were extracted. It is not necessary to report all the procedures in the methodology. I suggest only mentioning the manuals or authors.

I also recommend reporting the type of soil (classification) you worked on and its physical and chemical properties.

Results

Report the results based on the variables studied. The results are reported in a very descriptive and not very analytical manner. It is not recommended to report data again in the manuscript if they are already in figures or tables.

Discussion

The problem the authors want to solve needs to be clearly stated, and the discussion is based more on comparing their results with those of other authors. In addition, there is a section on the importance of cladiolus for human and livestock nutrition.

Conclusions

The Opuntia stricta cladodes with a length of 16-20 centimeters (cm) present greater physical and physical-chemical attributes when submitted to a range biofertilizer doses. The biofertilizer dose of 20% promotes increase in the contents of soluble solids, phenolic compounds, and total sugars. Nevertheless, it did not affect the content of pigments. The cladodes 4-8 cm long grown with and without biofertilizers presented greater respiratory rates.

Under Mexico's inhabitants' standards, the vegetable cladodes (4-8 cm) would have the lowest production costs but are preferred as a vegetable. According to the conclusions, large cladioles improve their quality with organic fertilization, but their use would only be for livestock and would raise production costs. The information on the use and taste of cladiolus in Brazil needs to be reported in the manuscript. It would be worthwhile to consult producers and the population.

Author Response

Dear authors: 

After reading the manuscript “Physical and chemical characteristics of young cladodes of Opuntia stricta (Haw.) Haw. with doses of bio-fertilizer”, the comments and suggestions are the following.

  1. Change the title. The authors indicate that they worked with bio-fertilizers but have not mentioned what kind of microorganisms were developed with the fermentation of bovine manure. It is more appropriate to use organic fertilization.

The change of title was accepted. The authors are grateful for the suggestion.

Organic fertilization with biofertilizer changes the physical and chemical characteristics of young cladodes of Opuntia stricta (Haw.) Haw.

  1. In Mexico, organic fertilization and irrigation are widely used in the production of cladiolus for human consumption, preferably small in size. However, for livestock consumption, they are not fertilized, and cladioles are allowed to grow up to 20 cm. These are more fibrous and may be of higher quality in terms of sugars, but they are not palatable.

In general, in Brazilian cuisine, there is no consumption of palm cladodes. In some states of Brazil, a small trade for palm cladodes is observed, especially in small urban fairs. Thus, this work is an initiative to characterize the post-harvest quality regarding the development (size) of palm cladodes from an organic fertilizer source (biofertilizer), intended for human consumption. Therefore, it is a commercial alternative for small and medium palm producers in the Brazilian semi-arid region. The cladodes in culture conditions presented also highlight the balance regarding palatability (SS/TA ratio) for the highest development stage with the studied species.

  1. In the literature review, the authors concentrated on research carried out in their country but did not carry out a much more detailed analysis of cladiolus and the various variables to be studied that have been carried out in other countries. Furthermore, they need to indicate the management of Opuntia in their country, the problems they have, production costs, and even acceptance by the population in terms of size. In such a way that they do not justify the study or the problem they want to solve (is there a problem with the size of the cladiolus, fertilization, production costs they want to lower, etc.), that would allow them to put forward hypotheses more related to the research. 

Following the reviewer's comment, a paragraph has been included in the introduction that addresses the information listed above. The part of the text is following next: “Its use in human food is widespread in some parts of the world, such as Mexico [4]. In Brazil, the use of young Opuntia spp. cladodes is still recent, requiring the development of technologies for the production of cladodes with nutritional quality. In general, its handling is done in a rustic way, with organic fertilization being a viable alternative for producers, reducing production costs and improving the nutritional qualities of cladia intended for human consumption [1]”.

Point 4.3 was introduced to work, addressing a little more about how to manage and choose cladodes, using works from Brazil and Mexico on L493-565 which have added citations [43] through [54]:

4.3. Management and ideal size of cladodes for consumption

Generally in Brazil, cactus is used for animal feed, making its management carried out with low-intensity and few technologies being used by producers. This is true to not only during planting but also later during the growth and development phases of the crop. Normally, cactus producers in the northeast region of Brazil make use of organic fertilizers such as bovine, goat and poultry manure, which are added in furrows or holes during planting. The use of organic amendments in agricultural systems for cactus has been widespread, given their important economic and environmental contributions [43].

From an economic point of view, cactus producers can realize increased income due to the decrease in use of industrialized fertilizers. This makes cactus cultivation possible for less capitalized farmers, which can generate promising results from such mineral fertilization [43].

Other studies verify benefits to using organic fertilizers during cactus production. For example, researchers have found that cactus responds well to organic fertilization since soils in semi-arid region tend to have low organic matter content. Therefore, such organic fertilization with livestock manure can increase organic matter in these soils and be essential to crop production [44]. It is recommended to fertilize cactus with bovine and goat manure with 10 to 30 metric tons /hectare during planting and then every two years during the beginning of the rainy season [45]. The presence of animal manure, mainly from cattle, on most agricultural properties can help maintain soil fertility Brazil’s semi-arid regions [46].

Few studies have focused on the cultural management of Opuntia spp. cactus in Brazil. Most research has focused on the use of cactus cladodes in human food, as well as for the manufacture of differentiated, value-added products [29]. Research is needed to determining the ideal size of cactus cladodes for human consumption.

In general in Brazilian cuisine, there is no consumption of cactus cladodes. In some states of Brazil, there is a small trade in such cladodes, mainly in small urban fairs. Thus, this work is a first step in characterizing the post-harvest quality and size of Opuntia spp.  cladodes intended for human consumption using organic fertilizer. Our research results suggest that cactus production is a viable commercial alternative for small- and medium-sized cactus producers in Brazil’s semi-arid regions. It is important to for cactus cladodes to be balanced in terms of palatability measured as the soluble solid and titratable acidity (SS/TA) ratio for the more mature developmental stages of the crop.

Likewise, it was observed for the 'Redonda' and 'Gigante' cactus cultivars that yield, physical attributes, and bioactive compounds of the cactus buds increased with developmental stage advancement up to 20 centimeters which is the ideal for consumption [29]. This same study also reported that yield varied according to the size of the cactus cladodes and the number of thorns [29]. Cactus yield for food can play an important role when selecting the best sized shoots to optimize propagation for both consumption and agro-industrial processing.

In recent years in Brazil, some authors have determined that the cactus cladode size of 15 to 20 centimeters as ideal for human consumption. Cladodes should be harvested 30 to 60 days after sprouting and weigh 80 to 120 grams. They must also be tender, young, fine, fresh in appearance, turgid and bright green in color so that they can also be used in various culinary recipes from around the world. [29]. In Mexico, researchers have determined that the ideal size of cactus cladodes for human food is 10 to 12 centimeters (cm) in length and that the cladodes need to be tender [47]. Another study concluded that ideal cladodes were 20 cm long and weighing 90 to 100 grams [48]. Cladodes to be exported should be between 17 and 21 cm in length and those selected for national consumption should be between 21 and 25 cm [49]. While [50,51] state that cladodes are ready for consumption when they are between 7 and 30 cm long.

Mexico is one of the origin centers of Opuntia spp. where consumption of cladodes is ancestral [52]. Cladodes are consumed as vegetables. Harvested area has more than doubled from 5269 to 12,105 ha between 1990 and 2012 [49]. Cactus plants, due to their adaptation to moisture deficit and semi-desert and desert climates, are a food source with great potential for the development of plantations [53]. Mexico is the main producer of cactus cladodes in the world at 74% of global production and is the largest consumer of both fresh and processed cactus. However, other markets such as those in the United States and Canada present a growing opportunity for export [49]. Due to their nutraceutical characteristics, cactus cladodes have shown great interest in the world market [52], as they are beneficial for various treatments [49]. Furthermore, they contain vitamin C, minerals, as well as soluble and insoluble fiber [49,52].

For the commercial use of cactus cladodes in Mexico, producers already use commercial size standards. However, it is important to follow similar commercial standards for export and national trade. These standards do not yet exist in Brazil since consumption of cactus cladodes in Brazil is still regional and limited. Further research and incentives from governments and public entities are needed in order to expand the production and consumption of cactus as a vegetable. Brazil has great potential is developing cactus as an export crop since it is tied with Tunisia in global production (600,000 hectares for each country) with Mexico at third with 230,000 hectares of cultivated cactus [54]. This holds great importance not only with Mexico but also with other countries and markets around the world.

  1. The hypothesis is that the "increase of the biofertilizer's availability is directly proportional to the physical and physical-chemical qualities of the Opuntia stricta regardless of the growth stage evaluated. However, as the research question or the contribution to the generation of knowledge on Opuntia production needs to be established, the hypothesis is not a tentative answer.

The authors recognize the mistake of the presented hypothesis. Thus, the hypothesis was modified for better understanding.

  1. Objectives: to evaluate the physical and physical-chemical qualities of different lengths of Opuntia stricta cladodes grown with different doses of biofertilizer. They further add that the objectives are for:
  2. a) low-cost organic matter (e.g., biofertilizer) with a direct effect on the physicochemical quality of the cladodes for fresh consumption as an unconventional vegetable.
  3. b) Cultivation techniques that can improve cladode and fruit quality are practices which are decisive in the marketing of crop yield.
  4. c) To improve the market value of the crop and the economic income of farmers and rural populations in arid and semi-arid regions in Brazil and elsewhere in the world.

The authors did not assess, nor did they indicate, the current costs of cladiolus production and how much these costs would be reduced by applying organic fertilizers or bio fertilizers. Furthermore, they do not indicate or survey consumer opinion on cladiolus size, let alone mention how to improve the market value of the crop and the economic income of farmers and rural populations.

The authors recognize the mistake regarding the objective of the work. In this way, only the initial objective a) was maintained, and description of b) and c) were removed.

Materials and Methods

It is recommended to separate into subheadings:

Site location and Laboratory analysis. The laboratory analyses recommend subdividing Physical-chemical analyses, Nutritional characteristics, respiratory rate, and Statistical analyses).

Following the suggestion, the methodology was separated into subheadings.

It is also recommended to indicate the manuals or authors from which the procedures of each section were extracted. It is not necessary to report all the procedures in the methodology. I suggest only mentioning the manuals or authors.

I also recommend reporting the type of soil (classification) you worked on and its physical and chemical properties.

The authors are grateful for the suggestions and they were accepted.

The figure was added to the material and methods to improve the characterization.

Results

Report the results based on the variables studied. The results are reported in a very descriptive and not very analytical manner. It is not recommended to report data again in the manuscript if they are already in figures or tables.

Following the suggestions, changes were made in the text.

Discussion

The problem the authors want to solve needs to be clearly stated, and the discussion is based more on comparing their results with those of other authors. In addition, there is a section on the importance of cladiolus for human and livestock nutrition.

According to the comment, authors made some changes in the discussions.

Conclusions

The Opuntia stricta cladodes with a length of 16-20 centimeters (cm) present greater physical and physical-chemical attributes when submitted to a range biofertilizer doses. The biofertilizer dose of 20% promotes increase in the contents of soluble solids, phenolic compounds, and total sugars. Nevertheless, it did not affect the content of pigments. The cladodes 4-8 cm long grown with and without biofertilizers presented greater respiratory rates.

Under Mexico's inhabitants' standards, the vegetable cladodes (4-8 cm) would have the lowest production costs but are preferred as a vegetable. According to the conclusions, large cladioles improve their quality with organic fertilization, but their use would only be for livestock and would raise production costs. The information on the use and taste of cladiolus in Brazil needs to be reported in the manuscript. It would be worthwhile to consult producers and the population.

According to the comment, changes have been made in the conclusions.

Reviewer 2 Report

Please attend to the comments in the document.
Eliminate or adjust in the introduction section 4.2

Author Response

Dear authors: 

After reading the manuscript “Physical and chemical characteristics of young cladodes of Opuntia stricta (Haw.) Haw. with doses of bio-fertilizer”, the comments and suggestions are the following.

Please pay attention to the comments in the document.

Delete or adjust in the introduction section 4.2

We have made all edits outlined in the pdf version with comments as well as outlined here.

Abstract

The suggestion was accepted.

Material and methods

Following the reviewer's suggestion, the second paragraph of topic 2.1 was rewritten.

The Table 1 was removed as suggested. Thus, the tables that were 2 and 3 before, became 1 and 2.

The type of soil used in the experiment was included as requested.

Following the recommendations, paragraphs 1 and 2 of topic 2.2 were rewritten.

Results

Following the suggestions, improvements were made in the figures. In addition, figures 7a and 7b were merged, as requested.

Following the reviewer's suggestion, the text of item 4.2 was adjusted. There was the readjustment of references 36, 37, 38 and 39.

Round 2

Reviewer 1 Report

Dear Editor: 

After reading the manuscript, it can be accepted for publication. It has greatly improved and is of more interest to Cladioli researchers worldwide. However, I have one comment. I recommend reporting the soil classification (Brazilian and international, Soil Taxonomy or WRB). If the authors only have the Brazilian soil classification, write the main characteristics.

Greeting 

Dra. Ma del Carmen Gutiérrez Castorena